# Small Intestinal Bacterial Overgrowth as Potential Therapeutic Target in Parkinson’s Disease

**DOI:** 10.3390/ijms222111663

**Published:** 2021-10-28

**Authors:** Adela Dănău, Laura Dumitrescu, Antonia Lefter, Delia Tulbă, Bogdan Ovidiu Popescu

**Affiliations:** 1Department of Clinical Neurosciences, Colentina Clinical Hospital, Neurology Division, Carol Davila University of Medicine and Pharmacy, 020021 Bucharest, Romania; adela.danau@drd.umfcd.ro (A.D.); laura.dumitrescu@drd.umfcd.ro (L.D.); antonia.lefter@rez.umfcd.ro (A.L.); delia.tulba@umfcd.ro (D.T.); 2Department of Clinical Neurosciences, Carol Davila University of Medicine and Pharmacy, 030167 Bucharest, Romania; 3Laboratory of Neurosciences and Experimental Myology, Victor Babeș National Institute of Pathology, 050096 Bucharest, Romania

**Keywords:** gut dysbiosis, gut–brain axis, microbiota, Parkinson’s disease, small intestinal bacterial overgrowth, SIBO

## Abstract

Increasing evidence suggests that the gut microbiota and the brain are closely connected via the so-called gut–brain axis. Small intestinal bacterial overgrowth (SIBO) is a gut dysbiosis in which the small intestine is abundantly colonized by bacteria that are typically found in the colon. Though not a disease, it may result in intestinal symptoms caused by the accumulation of microbial gases in the intestine. Intestinal inflammation, malabsorption and vitamin imbalances may also develop. SIBO can be eradicated by one or several courses of antibiotics but reappears if the predisposing condition persists. Parkinson’s disease (PD) is a common neurodegenerative proteinopathy for which disease modifying interventions are not available. Sporadic forms may start in the gut years before the development of clinical features. Increased gastrointestinal transit time is present in most people with PD early during the course of the disease, predisposing to gut dysbiosis, including SIBO. The role that gut dysbiosis may play in the etiopathogenesis of PD is not fully understood yet. Here, we discuss the possibility that SIBO could contribute to the progression of PD, by promoting or preventing neurodegeneration, thus being a potential target for treatments aiming at slowing down the progression of PD. The direct symptomatic impact of SIBO and its impact on symptomatic medication are also briefly discussed.

## 1. Introduction

Parkinson’s disease (PD) is a prevalent and highly disabling neurodegenerative disorder affecting more than 6 million people worldwide [1]. In addition to typical parkinsonism, which is considered the core clinical feature, PD has a variety of nonmotor manifestations, including hyposmia, dysautonomia (e.g., constipation, orthostatic hypotension, etc.), neurocognitive impairment and sleep disturbances, that significantly contribute to the overall disease burden [1,2,3,4,5]. Currently there are no therapeutic interventions that can prevent, delay, stop or reverse the progression of PD, but only symptomatic treatments, with net benefits that diminish as the disease advances (i.e., for parkinsonism mainly levodopa with a decarboxylase inhibitor, dopamine receptor agonists, catechol-O methyltransferase inhibitors, and monoamine oxidase inhibitors) [1,6].

Sporadic forms of PD are thought to result from complex interactions between different genetic and environmental factors [6,7,8,9]. From a pathological perspective, PD is mainly characterized by alpha-synuclein phosphorylation and misfolding resulting in amyloid formation with intraneuronal accumulation of insoluble aggregates (i.e., Lewy bodies and neuritis) and subsequent neurodegeneration, accompanied by neuroinflammatory changes [10,11]. Increasing evidence suggests that in most cases the pathology begins in the gut and/or in the olfactory mucosa, spreading to the brain in a prion-like fashion, along nerve trajectories [7,8,9,10,11,12,13,14,15,16]. Thus, the gut microbiota may play a potentially important role in the etiopathogenesis of sporadic PD [16,17,18,19,20,21,22].

The gut microbiota is a vast and complex ecosystem with high intra- and interindividual variability. It comprises of bacteria, archaea, fungi, viruses, and parasites (the latter completely eradicated in most people from developed countries) and has bidirectional connections with the brain, via the so-called gut–brain axis [23]. The crosstalk between the gut microbiota and the host is typically mutually beneficial; however, the structure and function of the microbiota is constantly changing, being susceptible to environmental factors, such as diet and antibiotic use, and alterations in its quantity or quality may promote the development of certain diseases, a state known as dysbiosis [23,24]. Gut dysbiosis is prevalent in PD but it is not clear if it precedes PD pathology or is a consequence of the gut dysmotility or other predisposing conditions related to PD [17,18,19,20,21,22]. Irrespective of the initial events, gut dysbiosis may lead to intestinal inflammation and barrier dysfunction (i.e., ‘leaky gut’), as well as blood–brain barrier alteration and an inflammatory shift of the brain milieu, the presence of these changes being increasingly documented in PD [25,26,27,28,29,30,31]. Additionally, gut dysbiosis might promote alpha-synuclein expression and aggregation in neurons of the enteric plexuses and other intestinal cells [32,33,34,35], may result in increased exposure to various microbial or nonmicrobial xenobiotics with direct or indirect proinflammatory or neurotoxic effects [23,24,36], and may cause nutritional imbalances (e.g., malabsorption, altered production of vitamins by the microbiota) [23,24], all these mechanisms potentially causing neuronal injury or interfering with the neuronal susceptibility to injury and therefore with the onset and progression of PD [37].

Small intestinal bacterial overgrowth (SIBO) is a gut dysbiosis in which the small intestine is excessively colonized by bacteria that are typically found in the large intestine [38]. It appears mainly in individuals that have predisposing conditions, such as decreased gastrointestinal motility / increased gastrointestinal transit time [38,39]. It may result in nonspecific symptoms, related mostly to intestinal distension caused by the gases produced by the excessive intestinal microbiota or by intestinal inflammation, but is not necessarily symptomatic and is not a disease [38,39]. SIBO-related symptoms typically disappear after one or several courses of antibiotics but recur if the predisposing condition persists [22,40,41]. Considering the potential immunomodulatory/proinflammatory effects of SIBO, its impact on intestinal barrier permeability and the potential effects on the levels of the microbial gases and other microbial products that are produced in the small intestine, the involvement of SIBO in the etiopathogeneses of neurodegenerative disorders seems plausible [42,43,44,45]. In this review we discuss the possibility that SIBO could contribute to the progression of PD, by promoting or preventing neurodegeneration, thus being a potential therapeutic target not only for symptomatic management (i.e., by improving gastrointestinal symptoms and increasing the efficacy of the oral medication used for the symptomatic treatment of PD) but also for delaying or stopping disease progression.

## 2. SIBO in Clinical Practice

The commonly accepted reference standard for defining SIBO is a bacterial count of at least 10^3^ colony forming units (CFU) per mL of small intestine fluid, in either aerobic or anaerobic conditions [38,39,46,47,48,49,50]. There is no correlation between small intestine pathology and SIBO, and the recommended 10^3^ UFC/mL cut-off was chosen because the concentration of plankton bacteria in the small intestine is typically lower than that [46,49,50,51,52,53,54]. Asymptomatic individuals may have small intestine bacterial counts as high as 10^5^ UFC/mL, in the absence of any known predisposing conditions, therefore some authors suggest this would be a better cut-off [55,56]. Additionally, the reference range of the ‘healthy’ gut microbiota for each topographic niche is still under debate and the products of the microbiota (not only its count or composition) are also relevant for its impact on health, therefore the above definition for SIBO is not a true gold standard [38,46,57].

The collection of intestinal fluid required for the bacterial count is performed by endoscopy (an invasive procedure, albeit minimal, that is more resource-consuming), therefore carbohydrate breath tests are increasingly recommended and used in clinical practice for identifying SIBO in people with predisposing conditions and persistent symptoms of unknown cause [38,46]. These tests are useful for the evaluation of several other common gastrointestinal conditions, are non-invasive, relatively inexpensive, widely available, and safe [38,46,55]. Moreover, unlike the small intestine cultures, breath tests offer both a quantitative and a qualitative assessment of the small intestine microbiota, the microbial gases predominantly produced in the context of SIBO having potentially different implications in health and disease [38,46]. The rationale behind performing breath tests for SIBO (see Table 1) is that bacteria, archaea, and fungi residing in the digestive tract metabolize simple carbohydrates to gases which are not produced by human cells; these gases easily pass into the blood stream and are subsequently exhaled. The most useful substrates for identifying SIBO are glucose, metabolized by the intestinal microbiota (mostly bacteria) to hydrogen (H_2_), and lactulose, metabolized to H_2_ and methane [46,47]. The recommended cut-off value for H_2_ is a rise of at least 20 ppm by 90 min following 75 g glucose or 10 g lactulose, administered orally, à jeun [46,55]. The cut-off level for methane is a level of at least 10 ppm, any time during the test (i.e., at baseline, à jeun, or up to 2 h following the ingestion of 10 g lactulose) [46,55]. Using 10^3^ CFU/mL as reference, the positive and negative predictive values for the above tests is around 70%, increasing when both H_2_ and methane are tested [38,46,55].

Although it is not a disease, SIBO can cause intestinal symptoms related to the accumulation of microbial gases in the small intestine, triggered or exacerbated several minutes to hours after a meal (e.g., bloating, belching, flatulence, abdominal pain or discomfort, etc.), as well as changes in the intestinal transit times (i.e., diarrhea or, more rarely, constipation), and altered characteristics of the stool (i.e., consistency, shape, color, smell) [38,39,58,59]. Severe SIBO can also lead to macronutrient malabsorption (carbohydrate, fat, proteins) resulting in weight loss, as well as to vitamin imbalances (i.e., deficiency or excess, especially in relation to vitamins produced by the microbiota and lipid-soluble vitamins, such as vitamin D), while in rare instances it may also cause intestinal inflammation (e.g., ileitis) resulting in abdominal pain, decreased intestinal transit time, bloody stools, etc. [38,39,60,61]. Thus, SIBO can be suspected in all individuals with predisposing conditions, especially if they have bloating and/or other unexplained intestinal symptoms, unexplained weight loss, macronutrient malabsorption and/or vitamin imbalances [38,58,59].

Several patterns of H_2_ and methane production have been described in SIBO, mainly: hydrogen-predominant, methane-predominant, and mixed H_2_/methane bacterial overgrowth, the first two being the most studied [38,46,47]. The patients with either pattern may share similar symptoms, but those with excessive methane production are about five times more likely to experience constipation, its severity directly correlating with the methane levels (i.e., evidence suggests that methane may decrease the motility of the colon); moreover, the efficiency of different antibiotic regimes seems to differ between these groups, making the distinction clinically relevant [38,46,47,55].

## 3. SIBO and PD

SIBO is relatively common in people with PD, including those with recent onset of motor symptoms, around half of the patients with PD testing positive for SIBO, compared with only up to a quarter in the general population [42,43,44,45]. Notably, however, the prevalence may be as low as 14%, or as high as 67%, depending on the demographic and clinical characteristic of the population included in the study, as well as on the testing method that has been used (i.e., small intestine fluid bacterial count, H_2_ glucose breath testing, or H_2_ and methane lactulose breath testing, the highest percentages being reported in studies using both glucose and lactulose H_2_ and methane breath testing and the lowest in studies using H_2_ glucose testing alone) [43,62,63,64,65,66]. Additional conditions and medications interfering with the gut microbiota also need to be excluded for a reliable account of the direct association between SIBO and PD [45,62]. Despite this variability, a recently published meta-analysis found a strong association between SIBO and PD when compared to healthy controls, with a pooled prevalence of SIBO in patients with PD of 47% (95% confidence interval 36–56), higher in Western countries (i.e., 52%, compared to 33% in Eastern countries), and an overall odds ratio of SIBO in patients with PD of 5.22 (95% confidence interval 3.33–8.19, *p* < 0.00001) [45].

The relation between SIBO and specific characteristics of sporadic PD is not straightforward, but it is biologically plausible that SIBO might influence the etiopathogenesis, clinical phenotype and progression of sporadic PD, with potentially different effects of H_2_-predominant versus methane-predominant SIBO [17,25,26,27,28,29,30,37,62,65]. Furthermore, it is plausible that SIBO may interfere with the bioavailability and absorption of enterically administered medication used for the symptomatic treatment of PD, and increase or decrease gut motility, thus potentially exacerbating or alleviating PD-related intestinal symptoms [44,67,68,69]. In respect to its involvement in the etiopathogenesis of PD, SIBO might induce a local inflammatory response that disrupts the intestinal barrier integrity by affecting tight junctions and subsequently increases intestinal permeability, which may favor exposure of the intestinal mucosa to bacterial exotoxins, such as lipopolysaccharide (LPS) [37,70,71]. LPS and other products of the gut microbiota ascribed to local SIBO-related inflammatory changes may trigger and/or enhance alpha-synuclein amyloidogenesis and along with other xenobiotic compounds may increase the neuronal susceptibility to neurodegeneration by direct or indirect mechanisms [17,27,28,29,30,37]. Additionally, SIBO could contribute to nutritional imbalances that may increase neuronal susceptibility to injury [25,26,27,28,29,37,62,65]. On the other hand, the pathophysiology of SIBO in PD probably entails the early impairment of gastrointestinal motility that occurs in people with PD, due to the involvement of the enteric nervous system as well as of the autonomic nervous system, especially the vagus nerve nuclei and vagus nerve, which innervates the stomach, the small and large intestines, and the appendix [10]. All these are summarized in Figure 1.

There have been inconsistent findings regarding the association of SIBO with more advanced PD or longer disease duration [42,43,44,45]. The above-mentioned meta-analysis failed to identify any statistically significant differences of SIBO prevalence in people with PD and constipation, bloating, diarrhea, or longer disease duration [45]. However, H_2_-predominant versus methane-predominant SIBO may have different gastrointestinal effects, canceling out statistical significance when SIBO is analyzed overall, and people with PD typically already have increased orocecal transit times and constipation which could be improved to a certain degree by SIBO in some individuals [43]. Different studies found that the presence of SIBO in people with PD is associated with the severity of parkinsonism but not necessarily with the severity of levodopa-related motor complications (i.e., motor fluctuations and dyskinesia), while others found that people with SIBO and PD have worse motor fluctuations, more specifically longer daily off times and increased frequency of delayed on [43,45]. Moreover, improvement of motor fluctuations has been achieved after SIBO eradication [43,63,72], indirect evidence suggesting that *Enterococcus* species that overpopulate the small intestine in people with SIBO and express decarboxylases could metabolize levodopa prior to its absorption [68,69]. Concerning the severity of the intestinal symptoms, the findings are also inconclusive, but milder intestinal symptoms, especially constipation, have been reported in patients with PD and SIBO [44]. Though weight loss is a common occurrence in advanced PD, no correlation between SIBO and body weight or weight loss was found [62].

Available evidence suggests that medication used for the symptomatic improvement of parkinsonism in people with PD, especially catechol-O methyltransferase inhibitors, may also interfere with the composition of the gut microbiota [19,73]. Similarly, the PD medication could potentially interfere with the risk of developing SIBO, for example by increasing the gastrointestinal transit times or by facilitating bacterial colonization (i.e., some bacteria produce and use dopamine) [73,74]. In this respect, the available studies found no correlation between PD medication and SIBO, and the eradication of SIBO did not affect the pharmacokinetics of levodopa, despite improving motor fluctuations [43]. However, the available epidemiological data are scarce, and effect of levodopa-based products (as well as that of other drugs) might be difficult to assess, since almost all the patients included in the studies are treated. It is plausible to consider that the risk of SIBO could be higher in people with advanced PD treated with levodopa-based products administered by continuous intestinal infusion (e.g., levodopa/carbidopa intestinal gel), but data are lacking.

## 4. Molecular Hydrogen and PD

H_2_ is a bioactive gas utilized both by human and microbial cells [75,76]. In humans, H_2_ is exclusively produced by the microbiota (e.g., *Blautia* spp., *Clostridium* spp., etc.), mainly within the gut, via food fermentation—i.e., carbohydrate breakdown, more specifically glycolysis and acetate formation; small amounts are also produced by bacteria outside the gut [43,63,75,77]. H_2_ has a key role in many microbial metabolic pathways and is used as nutrient by certain bacteria and archaea that are called hydrogenotrophs or H_2_ cross-feeders, namely sulfate-reducing bacteria, acetogenic bacteria, and methanogenic archaea, that convert H_2_ into hydrogen sulfide, acetate, and methane, respectively [75]. Therefore, H_2_ modulates the composition and functionality of the microbiota, mainly by reinforcing hydrogen sulfide- and acetate-producing bacteria, and methane-producing archaea [75]. The relation between SIBO-related H_2_ and methane (over)production as well as the general mechanisms by which SIBO could contribute to the progression of sporadic PD are summarized in Figure 2.

Oxidative stress represents the imbalance between the production and elimination of reactive oxygen species, leading to excessive oxidation reactions, with increased excitotoxicity and mitochondrial dysfunction, resulting in neuronal damage, even apoptosis [78]. This makes oxidative stress one of the main pathogenetic players in neurodegenerative disorders, including PD [78,79]. A recent experimental study by Musgrove et al. [79] found that oxidative stress increases the production of oxidatively-modified alpha-synuclein (including nitrated alpha-synuclein), promotes its pathological aggregation and exacerbates neuronal loss in the dorsal motor nucleus of the vagus nerve [79]. Furthermore, the study found that nitrated forms of alpha-synuclein are highly transferable, excessive oxidative stress enhancing the interneuronal alfa-synuclein transfer, therefore facilitating the spread of alpha-synuclein pathology [79]. At cellular levels, H_2_ acts as an electron sink, that can accept a new bond or a lone pair of electrons, being part of many chemical reactions, mainly mitigating oxidative stress by neutralizing hydroxyl radicals [43,63,75,76,77]. Considering the involvement of oxidative stress in the pathogenesis of PD and the antioxidant properties of H_2_, the presence of lower amounts of H_2_ may have a negative impact on PD progression [63,76,77,78].

Increasing evidence suggests that the gut microbiota of people with PD produces lower net levels of H_2_ [63,80,81]. However, data on the prevalence of methane-predominant versus H_2_-predominant SIBO in people with PD is not available yet [43,45,63]. The amount of H_2_ produced by the microbiota depends on its composition and metabolism [81]. The greatest H_2_ producers are *Firmicutes*, a phylum that has been shown to be altered both in terms of relative abundance and composition in people with PD [20,21,75,81]. A simulation of the gut H_2_ production based on the previously reported microbiota composition found a more than 2-fold reduction in the amount of produced H_2_ in patients with PD compared with controls [80]. Interestingly, a recent study on 20 patients with PD and 20 healthy controls found that the gut microbiota of PD had significantly higher levels of *Desulfovibrio*, a hydrogen sulfide producing hydrogenotrophic bacteria, the authors speculating that the LPS, hydrogen sulfide and magnetite produced by the strains of the *Desulfovibrio* bacteria could trigger alpha-synuclein conformational changes and aggregation [82]. Concurrently, hydrogen sulfide was shown to have both beneficial (i.e., neuroprotection, increase in neurogenesis) and deleterious effects in animal models of PD [82,83]. The potential impact of methanogen archaea is discussed below.

Several PD animal model studies found evidence that enterally administered H_2_ may be neuroprotective, downregulating peripheral inflammation, neutralizing toxic hydroxyl radicals, reducing oxidative stress within the brain, and preserving cerebral vascular reactivity [43,63,77,84,85]. In this respect, Fu Y. et al. [77] found a neuroprotective effect of H_2_-enriched drinking water in a mouse model of PD, with a reduction of dopamine neurons loss by 16% compared to the control group [77]. Another hemi-parkinsonism mouse model study found beneficial effects of a Si-based agent that generates large amounts of H_2_, probably by reducing oxidative stress [86].

The evidence on the potential effect of H_2_ in people with PD is scarce, but a few trials have been conducted. A 48-week pilot randomized placebo-controlled double-blind trial in humans found that H_2_-enhanced water is safe and significantly improves motor scores in patients with PD treated with levodopa [87]. However, a subsequent multicenter randomized placebo-controlled double-blind clinical trial failed to identify any beneficial effects in patients with PD [88]. Another more recent 16-week randomized placebo-controlled double-blind study found that H_2_ gas inhalation is safe but has no beneficial effects in people with PD [89].

## 5. Gut Methanogenesis and Possible Links to PD

Methanogens are anaerobic microorganisms that produce methane as a by-product of the metabolization of simple substrates such as H_2_ and carbon dioxide [75,81,90]. During the last five decades, different methanogen species have been identified in a variety of human biological specimens, such as periodontal, intestinal, colonic, or vaginal samples, feces, and even brain abscesses [90]. Despite initially being considered anaerobic bacteria, all currently known methanogens are Archaea species, minor and less-known constituents of the human microbiome, which are obligate bacteria cross-feeders and interact with bacteria in syntrophic ways [81,90]. A methanogenic archaeome is present in protists, plants and animals, where it interacts with the hosts in a mutually beneficial fashion (i.e., symbiosis) [90]. The most prevalent species reported in humans are *Methanobrevibacter smithii* [91] and Ca. *Mmc. intestinalis* [92], hydrogenotrophs found in up to 95% of the samples [75,81]. The archaeoma present in the gastrointestinal system of mammals differs consistently from the environmental one, probably because of alteration by horizontal gene transfer over time [93].

Considering that archaea are unable to degrade sugar and are obligate bacteria cross-feeders, the quantity, composition, and function of the methanogenic archaeoma is subjected to change in the context of bacterial dysbiosis, such as SIBO [55,81]. On the other hand, methane is produced by archaea, which may have distinct clinical implications, and its excessive production seems to occur throughout the intestinal tract in individuals with small intestine overproduction, therefore the latest SIBO Guideline of the American College of Gastroenterology proposed a more accurate delineation between SIBO and methane overproduction, defining a new entity called intestinal methanogen overgrowth (IMO)—a terminology that will probably replace the current one, i.e., methane-predominant SIBO [55]. 

The impact of methanogens on human health and disease is still largely unknown. Based on the current knowledge, methanogens are thought to have local and remote modulatory effects on the immune system and barrier role against pathogenic microorganisms in the digestive tract [90]. They seem to be able to interact with the immune system at the gastrointestinal system level, since exposure of dendritic cells to *M. stadtmanae* and *M. smithii* leads to release of proinflammatory cytokines, up-regulation of the cell-surface receptors CD86 and CD197 and alteration of antimicrobial peptides gene expression [94]. Moreover, *M. stadtmanae* was reported to activate innate immune receptors, its recognition being mediated by TLR7 and TLR8 receptors with subsequent NLRP3 inflammasome activation [95]. Although there are conflicting data regarding their pathogenic role, methanogenic archaea are widely considered possible contributors to a large array of disorders, such as inflammatory bowel disease, colon cancer, diabetes, and obesity (reviewed in [96]).

As already mentioned, sporadic PD is a multifactorial disorder, characterized at a molecular level by abnormal aggregation of proteins into Lewy bodies in specific neuronal populations, with a central nervous system ascending temporal pattern in most cases, according to the Braak scenario [7,10,11]. The onset of the pathology of most PD cases seems to start in the gut, more precisely in the neuroenteric plexus, which suggests that sporadic PD might have a gastrointestinal trigger [78]. A few reports link the methanogenic archaea to pathogenic gastrointestinal mechanisms of PD. For instance, the archaeal 20S proteasome seems to be able to effectively proteolyze aggregated misfolded proteins, such as alpha-synuclein, tau, or mutant superoxide-dismutase 1 [97]. Furthermore, experimental expression of archaea proteasome-activating nucleotidase in rod photoreceptors of mice leads to effective counteraction of misfolding retinopathy in Gγ1 knock-out mice, implying rescue from a protein-misfolding neurodegenerative disease [98]. Concurrently, the molecular chaperone prefoldin found in archaea can increase the generation of amyloid beta oligomers and decrease amyloid beta fibrils, meaning that it increases toxicity in an experimental setting [99].

## 6. SIBO Treatment as Therapeutic Opportunity for Interfering with PD Progression

It is generally agreed that symptomatic SIBO should be treated with one or several courses of antibiotics, accompanied by interventions aiming to eliminate the predisposing factors (e.g., using prokinetics in people with increased gastrointestinal transit times), the latter being a sine qua non condition in order to prevent recurrence [38,40,46]. The clinical relevance of asymptomatic SIBO remains unknown, but similar to other gut dysbiosis states, accumulating evidence suggests that it may have negative health consequences, e.g., by contributing to an overall proinflammatory state, neurotoxicity, neurodegeneration, etc. [38,100].

As previously discussed, increased intestinal transit time is a common occurrence in early PD, appearing years before the onset of motor symptoms, and presumably explaining the higher prevalence of SIBO in this population [42,43,44,62,63]. Local inflammatory responses induced by SIBO or other gut dysbiosis may activate the immune system and disrupt the intestinal barrier permeability, allowing for a close interaction between alpha-synuclein and various bacterial products that may have amyloidogenic effects, triggering or enhancing alpha-synuclein misfolding and accumulation, key features of PD pathology [22,27,53,69]. Furthermore, an altered intestinal barrier grants access to the gut–brain axis for potentially proinflammatory and neurotoxic microbial and nonmicrobial compounds, that may increase neuronal susceptibility to neurodegeneration [22,27,70,71].

Among all microbial products, gases pass most easily through membranes and may interfere with cellular functions throughout the body. The composition of the gas produced by the small intestine microbiota in people with SIBO (which, as discussed, can be assessed in respect to its H_2_ and methane content after oral carbohydrate challenge with glucose and lactulose—see above) mainly depends on the concentration and the types of colonizing bacteria, both the structure and the function of the microbiota being important [38,39,46,47,48,49,50]. These gases may interfere directly with PD progression, both in positive and negative ways, H_2_ having potentially neuroprotective effects by diminishing inflammation and oxidative stress [43,63,77,84], and methane having either neuroprotective or neurodegeneration-promoting effects, by counteracting protein misfolding and removing aggregates of misfolded proteins, respectively, by increasing the aggregation of misfolded proteins [97,98,99]—see Table 2. Considering the above, specific SIBO-related mechanisms could serve as potential therapeutic interventions or targets for PD progression—see Figure 3. On the contrary, considering that SIBO may result in proinflammatory changes, both locally and within the brain, and may increase the permeability of the intestinal and blood–brain barriers, exposing neurons to potentially amyloidogenic and neurotoxic compounds and accelerating the progression of PD, its eradication should be considered in people with PD even in the absence of SIBO-related symptoms.

The management of symptomatic SIBO is centered on the use of oral antibiotics that successfully eradicate the bacteria overpopulating the small intestine; the recommended antibiotic regimens usually consist of rifaximin (which has the advantage of very limited systemic absorption), metronidazole, ciprofloxacin, doxycycline, neomycin or amoxicillin-clavulanate; the typical treatment duration is 7 to 14 days [22,40,41]. Evidence concerning the short- and long-term efficacy of one antibiotic over another is modest, so the antibiotic is chosen based on individual safety concerns, individual preferences and prior SIBO history [22,40,41]. Since treating the predisposing condition is essential for the long-term remission of SIBO, and SIBO in PD seems to be closely related with the increase in gastrointestinal transit times, administering antibiotics that also have prokinetic effects, such as azithromycin and erythromycin, in people with SIBO and PD is tempting—nevertheless safety and efficacy trials for SIBO in people with PD are lacking and their use in clinical practice is limited by theoretical safety concerns related to potential cardiovascular side effects [40]. Dietary changes and other symptomatic interventions can also be considered. The use of probiotics (i.e., live bacteria), especially *Lactobacillus* strands, may be useful in the management of SIBO, helping to repopulate the intestinal flora with health-promoting bacteria [22,40,41]. Prebiotics are another potentially useful intervention for SIBO, however evidence on their efficacy very limited [22,41].

Small studies found that SIBO eradication is safe and achievable on the short-term (i.e., up to 6 months) in people with PD, and that it may help improve motor fluctuations and possibly gastrointestinal symptoms [43,45]. Gut microbiota manipulation by probiotic use or fecal transplantation in people with PD may result in clinical improvement, especially concerning gastrointestinal symptoms [41,101]; however, the effect of this interventions in SIBO is unknown. Clinical trials regarding a potential disease modifying effect of these types of interventions in PD are lacking and the clinical relevance of the SIBO-related mechanisms that were shown to interfere with the progression of PD pathology in experimental settings is currently unknown. Since these types of interventions do not pose major safety concerns, further studies are warranted. Depending on the results of these studies, screening for SIBO and eradicating methane-predominant (or H_2_-predominant) SIBO in people with sporadic PD might become a cheap, safe, and accessible intervention that could be used in everyday clinical practice to help mitigate PD progression, especially in the early stages; SIBO eradication might also prove potentially useful in people with advanced PD, either by eliminating the direct impact of SIBO on motor and nonmotor symptoms or by changing the intestinal bioavailability of symptomatic medications. Moreover, if methane-predominant or H_2_-predominant SIBO prove to be risk factors for developing sporadic PD, SIBO screening and eradication could become the first intervention for preventing or delaying sporadic PD in people at risk. Concurrently, modulation of SIBO-related mechanisms, for example by administering H_2_-enhanced water [87,88], could help develop neuroprotective interventions aimed at preventing PD or slowing down its progression.

## 7. Conclusions

Up to this point, an important array of data regarding the human gut microbiota (archaeome included) has emerged, with possible implications for the pathogenesis of neurodegenerative disorders and their therapeutic targets. This line of development might be of particular relevance for sporadic PD, since the first pathological lesions seem to appear in the neurenteric plexuses, possibly as a consequence of local gastrointestinal processes, and then spread to the brain by a prion-like mechanism.

People with PD have higher incidence of SIBO, and SIBO may interfere with the progression of PD, both in negative and positive manners, depending on the levels of H_2_ and/or methane produced and possibly on other functional characteristics of the microorganisms colonizing the small intestine. The levels of H_2_ and methane produced by the intestinal microbiota can be estimated by measuring them in the expiratory air after a carbohydrate challenge, and could serve as variables in clinical trials that aim to assess the impact of SIBO in PD progression. Considering the above, SIBO eradication and other modulation/manipulation of the small intestine microbiota require further investigation as potential disease modifying interventions aiming at slowing down the progression of PD.

## Figures and Tables

**Figure 1 ijms-22-11663-f001:**
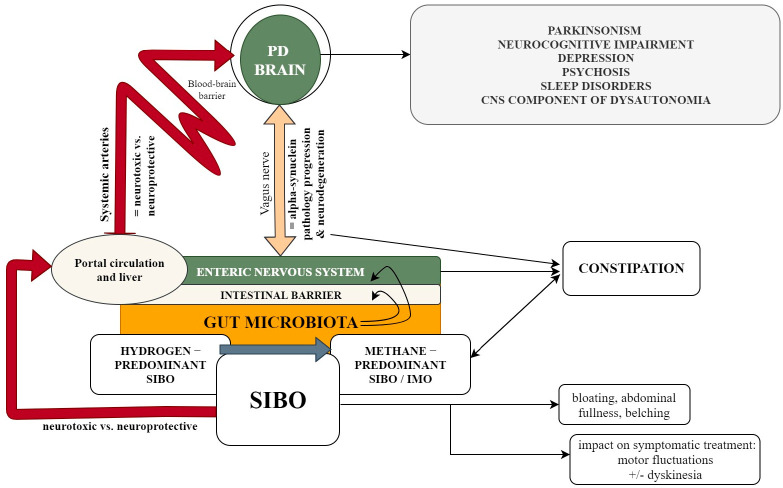
The relation between SIBO and specific characteristics of sporadic PD. The figure illustrates the potential relations between SIBO and specific clinical characteristics of sporadic PD, highlighting the potential pathogenic mechanisms that underlies these relations. SIBO may result in increased intestinal production of H_2_ and/or methane, with potential neuroprotective and neurotoxic consequences; H_2_ may also modulate the composition and function of the microbiota, for example by being substrate for hydrogenotrophic methanogenic archaea (see the main text), which may have potential indirect consequences on PD progression, both beneficial and deleterious. Gut dysbiosis, including SIBO, may alter the integrity of the intestinal barrier, thus exposing the central nervous system, via the neuronal and humoral pathways of the gut–brain axis, to microbial and non-microbial xenobiotic compounds that may have amyloidogenic or neurotoxic effects. On a clinical level SIBO may exacerbate (or alleviate) gastrointestinal symptoms related to PD, a potential connection existing between methane overproduction and the presence of constipation. SIBO may also interfere with drug bioavailability, interfering with their overall effect (i.e., symptomatic improvement, occurrence of side effects); this may be of particular importance in patients with advanced PD, that have motor fluctuations and dyskinesia.

**Figure 2 ijms-22-11663-f002:**
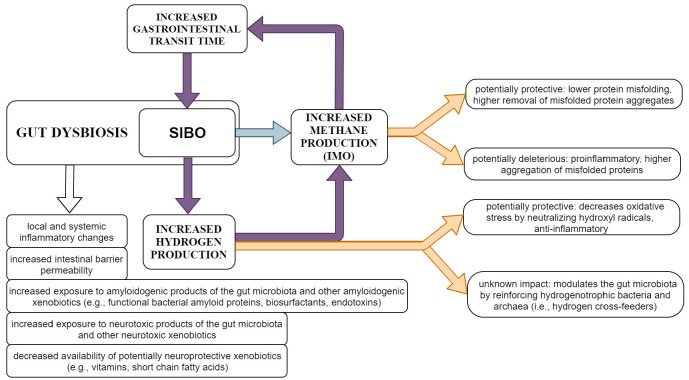
Main mechanisms underlying the relation between SIBO-related H_2_ and methane (over)production and the progression of sporadic PD. The figure illustrates the specific mechanisms that may link the intestinal production of H_2_ and methane with disease progression in people with sporadic PD and SIBO. General mechanisms related to gut dysbiosis, including SIBO, are also summarized. Depending on the functional characteristics of the SIBO microbiota (as well as particularities of the host) H_2_ and methane may have both beneficial and deleterious effects, requiring further investigation. Better understanding of these relations could offer means for personalized interventions and precision medicine in PD.

**Figure 3 ijms-22-11663-f003:**
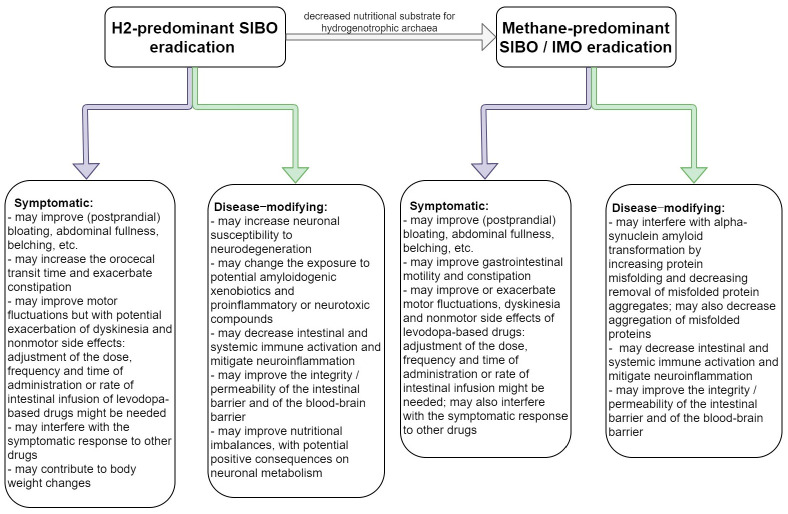
Expected effects of SIBO eradication in people with sporadic PD. The figure illustrates the potential symptomatic and disease-modifying consequences of SIBO eradication in people with sporadic PD. Noteworthy, interventions done for SIBO eradication would also interfere with the gastric and colonic gut microbiota. The effects of SIBO eradication and modulation require further investigation.

**Table 1 ijms-22-11663-t001:** Carbohydrate breath tests for SIBO.

Breath Test	Interpretation
**Glucose test** * (oral glucose, 75 g)	**SIBO (H2-predominant)** = raise in H_2_ levels of ≥20 ppm by 90 min
**Lactulose test** (oral lactulose, 10 g)	**H_2_-predominant SIBO** = raise in H_2_ levels of ≥20 ppm by 90 min **Methane-predominant SIBO** = methane level of ≥10 ppm, anytime for up to 2 h or at baseline **H_2_-/methane-predominant SIBO** = both of the above

* The glucose test does not evaluate methane production and may be false negative in those with increased gastric transit time, since glucose can be absorbed before reaching the small intestine [46,47].

**Table 2 ijms-22-11663-t002:** Possible roles of SIBO subtype in PD progression.

SIBO Subtype *	Presumptive Roles in PD Progression
H2-predominant SIBO	Protective: anti-inflammatory, decreases oxidative stress.Uncertain: modulates the gut microbiota (reinforces hydrogenotrophic bacteria and archaea); increased production of hydrogen sulfide.
Methane-predominant SIBO	Either protective or deleterious: may counteract protein misfolding and may effectively remove aggregates of misfolded proteins; proinflammatory and may increase aggregation of misfolded proteins.

* As per carbohydrate (i.e., lactulose) breath test.

## Data Availability

Not applicable.

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
