# Peer review of "Small Intestinal Bacterial Overgrowth as Potential Therapeutic Target in Parkinson’s Disease"

_ijms, 2021, doi:10.3390/ijms222111663_

Round 1
Reviewer 1 Report
This is an interesting review about the possible key role of “small intestinal bacterial overgrowth syndrome” (SIBO) as potential therapeutic target in Parkinson´s disease (PD). This work supports, based on an extensive review of data in the literature, that SIBO could contribute to the progression of sporadic forms of PD (promoting or preventing neurodegeneration), thus being a potential target for treatments aiming at slowing progression.
The subject matter is interesting and pertinent and moreover, the review is well structured, concise and easy to understand. However, in my view, several minor changes should be made to further clarify and improve the original before being accepted. Thus, the next issues should be addressed:
1) The manuscript needs English language editing and revision in order to correct some grammar and spelling mistakes.
2) Authors should cover the reference lists in an appropriate manner. There are many paragraphs (especially in the last one of the 1. Introduction) where relevant literature is not clearly and adequately justified.
3) In regard SIBO and PD (section 3), I recommend that the authors include a figure that serve as a model to summarize the main relation between SIBO and specific characteristics of sporadic PD.
4) The authors talked about gut metanogenesis and molecular hydrogen production in people with SIBO and their link with PD (sections 4 and 5). Due to the information is complex, I recommend the authors to summarize and clarify the main ideas of these sections in one or two figures, or in a mixed figure, to facilitate a better understanding to the readers.
5) Several studies demonstrate independent effects of PD medications on the microbiome (Hill-Burns et al., Mov Disord. 2017). How the variety of medication used to treat PD can affect SIBO, and therefore, its contribution to de progression of neurodegeneration in PD? The authors may discuss this briefly.
6) In their summary, the authors state that SIBO could contribute to the progression of PD, by promoting or preventing neurodegeneration, thus being a potential target for treatments aiming at slowing progression. How might this affect every day clinical practice? What action-specific consequences could result from the knowledge gained, in particular in earlies states of sporadic PD?
Author Response
We thank Reviewer 1 for their very helpful comments and recommendations.
We addressed all of the issues that were raised by Reviewer 1, as follows:
1) We reviewed the English language, corrected the spelling and grammar errors and rephrased a few parts to improve readability.
2) We revised the reference list and improved its relevance / accuracy.
3) We included a figure, summarizing the main relation between SIBO and specific characteristics of sporadic PD - see Figure 1.
4) We included a mixed figure, summarizing and clarifying the link between increased hydrogen and methane production in people with SIBO and sporadic PD, as well as the potential links between general mechanisms related to gut dysbiosis (including SIBO) and PD - see Figure 2.
5) We added a brief discussion about the impact that the medication used for the symptomatic treatment of PD-related parkinsonism could affect SIBO.
6) We added a brief comment on how SIBO management could affect everyday clinical practice and action-specific consequences in early stages and advanced disease. We also summarized in a figure the presumed consequences of eradicating hydrogen-predominant and methane-predominant SIBO in people with PD - see Figure 3.
Reviewer 2 Report
The review is well written and the topic is interesting. It deserves attention (SIBO - PD correlation is quite important).
I suggest to insert a chapter / paragraph about the chemical modifications (oxidation) that alpha-synuclein can undergo in oxidizing environments. This is an important and timing issue.
The authors only say that oxidative stress plays a fundamental role in PD, but they do not give any molecular information on what happens to the asyn molecule / structure / aggregation process.
A cartoon could also help: there are no figures but only tables. Thsi is proposed as a review , therefore the graphycal part is essential.
Author Response
We thank Reviewer 2 for their very helpful insight.
We addressed all of the recommendations made by Reviewer 2 as follows:
1) We included a paragraph about oxidative stress and PD and the effect of oxidative stress on alpha-synuclein (i.e., conformational changes, aggregation, interneuronal transmission).
2) We included 3 figures, summarizing and clarifying the main relation between SIBO and specific characteristics of sporadic PD, the link between increased hydrogen and methane production in people with SIBO and sporadic PD, and the potential consequences of eradicating hydrogen-predominant and methane-predominant SIBO in people with PD